# Privately Learning from Graphs with Applications in Fine-tuning Large Pretrained Models

**Haoteng Yin**
Purdue University
yinht@purdue.edu

**Rongzhe Wei, Eli Chien & Pan Li**
Georgia Institute of Technology
{rongzhe.wei,ichien6,panli}@gatech.edu

## ABSTRACT

Graphs offer unique insights into relationships and interactions between entities, complementing data modalities like text, images, and videos. By incorporating relational information from graph data, AI models can extend their capabilities beyond traditional tasks. However, relational data in sensitive domains such as finance and healthcare often contain private information, making privacy preservation crucial. Existing privacy-preserving methods, such as DP-SGD, which rely on gradient decoupling assumptions, are not well-suited for relational learning due to the inherent dependencies between coupled training samples. To address this challenge, we propose a privacy-preserving relational learning pipeline that decouples dependencies in sampled relations during training, ensuring differential privacy through a tailored application of DP-SGD. We apply this method to fine-tune large language models (LLMs) on sensitive graph data, and tackle the associated computational complexities. Our approach is evaluated on LLMs of varying sizes (e.g., BERT, Llama2) using real-world relational data from four text-attributed graphs. The results demonstrate significant improvements in relational learning tasks, all while maintaining robust privacy guarantees during training. Additionally, we explore the trade-offs between privacy, utility, and computational efficiency, offering insights into the practical deployment of our approach. Code is available at `https://github.com/Graph-COM/PvGaLM`.

## 1 INTRODUCTION

Graph data, commonly used to represent relationships between entities, are widely employed to model complex systems in the real world (Leskovec et al., 2007; Kwak et al., 2010; Shamsi et al., 2022; Madani et al., 2022). Recently, the relationships captured by graph structures have been used to enhance foundation models pretrained on other modalities (e.g., text and images) with complementary information, which enables these models to more effectively handle multi-entity tasks of emerging AI applications (Brown et al., 2020; Dosovitskiy et al., 2021; Zhang et al., 2024a; Madan et al., 2024). For instance, models trained on product descriptions or pictures may not fully capture the relationships revealed by user behaviors, such as co-purchases or co-viewings. Incorporating relational information can allow AI models to better meet users' needs (e.g., in product recommendations). Models pretrained on text or images and subsequently fine-tuned with relational information from graphs have recently been found applications in various domains (Ling et al., 2023), including healthcare (Wu et al., 2021; Zhang et al., 2022; Gao et al., 2023), finance (Ouyang et al., 2024), and computer vision (Li et al., 2023a). However, the relationships involved in these applications often contain sensitive information, such as patient-hospital visits for clinical diagnosis (Lu & Uddin, 2023), financial transactions for fraud detection (Kurshan & Shen, 2020), and social connections for recommendations (Zheng et al., 2022). This raises critical concerns about how to protect the privacy of relational data when exposed to AI models, motivating this research.

Differential Privacy (DP) (Dwork, 2006; Dwork et al., 2014) is widely considered the gold standard for measuring the privacy guarantees of data-processing algorithms (Xu et al., 2021; Pan et al., 2024). Current DP methods for model training, such as DP-SGD (Song et al., 2013; Abadi et al., 2016; Ponomareva et al., 2023), are primarily designed for tasks *other than* relational learning. DP-SGD, in particular, operates under the assumption that the gradient in each training step can be decoupled with respect to individual training samples that require privacy protection. Under this

assumption, DP-SGD controls the norm of the gradient induced by each sample, obfuscates it by adding Gaussian noise, and thus ensures a privacy guarantee. However, relational learning on graphs introduces unique challenges because each loss term typically involves multiple relationships (e.g., observed and unobserved relations from the graph), and each relationship involves multiple entities. Consequently, the gradient in relational learning cannot be decomposed into specific privacy-preserved samples, which violates the per-sample decoupling assumption, rendering DP-SGD not directly applicable.

Recent studies on privacy-preserving training of graph neural networks (GNNs) (Daigavane et al., 2021; Olatunji et al., 2023; Sajadmanesh & Gatica-Perez, 2021; Mueller et al., 2022; Sajadmanesh et al., 2023; Sajadmanesh & Gatica-Perez, 2024; Chien et al., 2024) do not address this issue, though they also work with relational data. These works are mainly designed for training with node classification labels, where the loss term is decomposable to specific nodes given the representations of these nodes output by GNNs. Their methods, which obfuscate the message-passing process to prevent privacy leakage during GNN encoding, *do not mitigate privacy risks arising from relational learning*, where the loss term cannot be decomposed on the supervision side. We further discuss the critical gap of current DP-GNNs in solving the problem of differentially private relational learning in Appx. A.1.

This study aims to introduce a privacy-preserving pipeline to address the gap in relational learning, where each loss term typically involves an observed relation paired with one or more unobserved relations for contrast. Common practice often couples the sampling of observed and unobserved relations, where removing or adding an observed relation may impact the gradients of multiple loss terms within the sampled batch, causing sensitive relational information leakage. Our key insight is to *decouple the sampling process* for observed and unobserved relations. By doing so, we ensure that removing or adding an observed relation affects at most one loss term, thereby limiting the sensitivity of data perturbation in relational learning and making it theoretically compatible with the privacy accounting of the DP-SGD framework.

As an application, we apply this privacy-preserving pipeline to fine-tune pretrained models using graph data, choosing LLMs as a proof of concept. Modern privacy libraries like Opacus (Yousefpour et al., 2021), TensorFlow Privacy (McMahan et al., 2018), and JAX Privacy (Balle et al., 2022) support clipping per-sample gradients to apply DP-SGD, but they are designed for each sample with a single input. The per-sample gradient computing relies on caching intermediate results of backpropagation, which are tracked through individual input tokens for sequence models (LLMs included) and leading to common inefficiencies. Relational learning introduces another dimension to the problem: Each loss term typically involves $K$ entities, and each entity with textual attributes contains $M$ tokens. Naively computing per-sample gradients requires keeping $\mathcal{O}(KM)$ gradient copies in memory per loss term, where modern GPUs easily run into out-of-memory issues for large models at moderate batch sizes. To eliminate the instantiation of $\mathcal{O}(KM)$ gradients, we propose to directly compute per-loss-term gradients by utilizing the low-rank structure of gradients through individual tokens. This approach alleviates memory constraints for models with billions of parameters, enabling efficient private fine-tuning on relational data with large batch sizes, which are empirically preferred to enhance privacy preservation (Li et al., 2021; Anil et al., 2022; Räisä et al., 2024).

We evaluate the proposed pipeline by testing whether target models can learn from private domains rich in relational data and generalize to new domains lacking such information to infer relationships between entities. Using real-world relational data from four text-attributed graphs, we fine-tune BERT (Devlin et al., 2019) and Llama2 (Touvron et al., 2023) at various model sizes (110M, 340M, 7B) under different levels of DP ($\epsilon \leq 10$) to stimulate two popular use cases that require data privacy: Cross-category co-purchase recommendation and Cross-region model deployment. Our results demonstrate that LLMs can effectively learn from relational data to address relational learning tasks, even with DP guarantees. The privacy risks of these fined-tuned models are empirically examined by conducting membership inference attacks. Additionally, we investigate the trade-offs between utility, privacy, and computational efficiency in LLM-based relational learning, extending existing research of privacy-preserving learning with LLMs on standard (non-relational) text data (Li et al., 2021). These findings offer valuable insights for the practical deployment of LLMs in privacy-preserving relational learning scenarios.

## 2  PRELIMINARIES: NOTATIONS AND STANDARD LEARNING VIA DP-SGD

Graph $(\mathcal{V}, \mathcal{E}, X)$ consists of a relation set $\mathcal{E}$ that describes the relationship between entities in $\mathcal{V} = [N]$. Each entity $v \in \mathcal{V}$ is associated with an attribute $X_v$ of text, images, or other data modalities.

$(\epsilon, \delta)$**-Differential Privacy.** A randomized mechanism $\mathcal{M}$ satisfies an $(\epsilon, \delta)$-differential privacy if for any adjacent datasets $\mathcal{D}, \mathcal{D}'$ that differ in one sample, and any output set $S \subset \mathrm{Range}(\mathcal{M})$, $\mathbb{P}(\mathcal{M}(\mathcal{D}) \in S) \le e^\epsilon \mathbb{P}(\mathcal{M}(\mathcal{D}') \in S) + \delta$, where $\epsilon, \delta \ge 0$ measure the privacy loss and smaller values imply stronger privacy guarantees. The notion of adjacent datasets can be generalized to relational data. Two relation sets $\mathcal{E}, \mathcal{E}'$ are considered adjacent if one can be obtained from the other by adding or removing a relation. We provide the formal definition of the DP guarantees provided by this work in Sec. 3.1.

**Standard DP Learning Paradigm.** Consider training a neural network using a mini-batch $\mathcal{B}$ of $b$ samples. The model parameters $\Theta$ is updated iteratively as $\Theta_{t+1} = \Theta_t - \eta g_t(\mathcal{B})$, where $\eta$ is the learning rate, and $\mathbf{g}_t(\mathcal{B}) = \partial\ell(\Theta_t; \mathcal{B})/\partial\Theta_t$ is the gradient of the loss $\ell$ on $\mathcal{B}$ w.r.t the parameters $\Theta_t$ at step $t$. Adding or removing one sample from $\mathcal{B}$ may change $\mathbf{g}(\mathcal{B})$, causing privacy leakage that can be measured by the *sensitivity* $\Delta_2 = \max_{\mathcal{B}, \mathcal{B}'} ||\mathbf{g}(\mathcal{B}) - \mathbf{g}(\mathcal{B}')||_2$, where $\mathcal{B}'$ and $\mathcal{B}$ are different in one sample $|(\mathcal{B}\backslash\mathcal{B}') \cup (\mathcal{B}'\backslash\mathcal{B})| = 1$.

DP-SGD (Song et al., 2013; Abadi et al., 2016) (see Alg. 1) was proposed to achieve data privacy for training deep learning models. It first clips *per-sample gradients* to control the sensitivity and then adds Gaussian noise to obfuscate the potential change as $\tilde{\mathbf{g}}(\mathcal{B}) = \frac{1}{b}\left[\sum_{x_i \in \mathcal{B}} \mathrm{Clip}(\mathbf{g}(x_i), C) + \mathcal{N}(0, \sigma^2 C^2 \mathbf{I})\right]$, where $\mathbf{g}(x_i)$ is the parameter gradient of the loss on sample $x_i$, and $\mathrm{Clip}(\mathbf{g}, C) = \mathbf{g}/\max(1, ||\mathbf{g}||_2/C)$ for some constant $C > 0$. At each step of parameter update, clipping per-sample gradients limits the sensitivity to at most $C$, and the Gaussian noise with standard deviation $\sigma C$ is added to achieve DP based on the Gaussian mechanism (Dwork et al., 2014). To obtain the DP guarantee for the entire training procedure, the composition theorem (Balle & Wang, 2018) is used to account for the total privacy loss over $T$ steps. Mini-batch sampling also allows for some privacy amplification, for which interested readers may check relevant works for more details (Balle et al., 2018; Wang et al., 2019).

## 3  METHODOLOGY

In this section, we first introduce the technical difficulty of applying standard DP-SGD when training models in relational learning. Then, we propose a pipeline that addresses this difficulty and can provably achieve differential privacy in learning from relational data. Lastly, we address the computing challenge induced by the control of gradient sensitivity involving multiple relations and entities, especially when applying the proposed pipeline to sequence models on text-attributed graphs.

**Enhance Models with Relational Data.**  Relationships provide complementary information to models trained on a specific modality, enabling them to more effectively handle tasks involving multiple entities. Suppose the representation of each entity $u$ is obtained from a model parameterized by $\Theta$ encoding its attribute, i.e., $\mathbf{h}_u = f_\Theta(X_u)$. A common approach of relational learning is to use relationships between entities to refine their representations (Yasunaga et al., 2022b; Duan et al., 2023; Xie et al., 2023). This is typically achieved via training based on a loss $\ell$ that can be generally written as the following form (Hadsell et al., 2006; Schroff et al., 2015; Song et al., 2016; Sohn, 2016; Ying et al., 2018; Oord et al., 2018). Given a tuple $E_i$, consisting of an observed (positive) relation $e_i^+ \in \mathcal{E}$ and several unobserved (negative) relations $\{e_{i_j}^-\}_{j=1}^k$ where $e_{i_j}^- \notin \mathcal{E}$, the loss is denoted as $\ell(\Theta; E_i)$. For a mini-batch $\mathcal{B}$ of tuples, the loss sum for $\mathcal{B}$ is computed as

$$\mathcal{L}_\Theta(\mathcal{B}) = \sum_{E_i \in \mathcal{B}} \ell(\Theta; E_i) = \sum_{E_i \in \mathcal{B}} \ell(\Theta; (e_i^+, \{e_{i_1}^-, \dots, e_{i_k}^-\})). \tag{1}$$

For convenience, let $\mathbf{z}_e = \Gamma(\mathbf{h}_u, \mathbf{h}_w)$ denote the joint representations of entities in each relationship. One popular choice of $\ell$ is the InfoNCE loss (Oord et al., 2018): $\ell(\Theta; E_i) = -\ln\left(\exp(\mathbf{z}_{e_i^+})/\sum_{e' \in E_i} \exp(\mathbf{z}_{e'})\right)$. Another choice is the pairwise Hinge loss $\ell(\Theta; E_i) = [\gamma + \mathbf{z}_{e_{i.}^-} - \mathbf{z}_{e_i^+}]_+$, which is commonly used for learning from complex multi-relations in knowledge

graphs (Bordes et al., 2013; Wang et al., 2014; Yang et al., 2015; Lin et al., 2015). Here, $\gamma$ represents the margin, and $\mathbf{z}_e$ also encodes the representation of the relationship besides the entities. Note that our method for relational learning can potentially be extended to the case where each relationship contains more than two entities, such as network motifs (Milo et al., 2002; Benson et al., 2016) and hyperedges (Berge, 1984), although the later discussion focuses on pairwise relationships.

### 3.1 Challenges in Privately Learning Relations

For relational learning, the information subjected to be protected is *the existence of a relation $e$* in the relation set $\mathcal{E}$, formally defined as follows.

**DP for Relational Data.** An $(\epsilon, \delta)$-DP algorithm for relational data ensures that the output obtained from a randomized mechanism $\mathcal{M} : \mathcal{X} \to \mathcal{Y}$ for any adjacent relation sets $\mathcal{E}, \mathcal{E}' \sim \mathcal{X}$ and measurable sets $Y \subset \mathcal{Y}$ satisfy: $\mathbb{P}(\mathcal{M}(\mathcal{E}) \in Y) \leq e^\epsilon \mathbb{P}(\mathcal{M}(\mathcal{E}') \in Y) + \delta$. Achieving DP for relational data limits the ability of the *best possible* adversary to uncover any specific relationship between entities used for training from the model parameters. When the set of relations is defined by a plain graph, the above concept reduces to edge-level DP, which is widely used in privacy-preserving graph algorithms (Hay et al., 2009).

Recall that DP-SGD relies on clipping per-sample gradients to control the sensitivity of the gradient sum of a mini-batch. In relational learning, the gradient sum $\mathbf{g}(\mathcal{B})$ of a mini-batch $\mathcal{B}$ is given by

$$\mathbf{g}(\mathcal{B}) = \frac{\partial \mathcal{L}_\Theta(\mathcal{B})}{\partial \Theta} = \sum_{E_i \in \mathcal{B}} \mathbf{g}(E_i) = \sum_{E_i \in \mathcal{B}} \left[ \underbrace{\frac{\partial \ell(\Theta; E_i)}{\partial \mathbf{z}_{e_i^+}} \cdot \frac{\partial \mathbf{z}_{e_i^+}}{\partial \Theta}}_{\text{Positive Relation}} + \underbrace{\sum_{j=1}^k \left( \frac{\partial \ell(\Theta; E_i)}{\partial \mathbf{z}_{e_{i_j}^-}} \cdot \frac{\partial \mathbf{z}_{e_{i_j}^-}}{\partial \Theta} \right)}_{\text{Negative Relations}} \right]. \quad (2)$$

The challenge comes from the fact that practical sampling of negative relations is usually coupled with positive relations in the same mini-batch. As a result, removing or adding a positive relation $e \in \mathcal{E}$ will not only change the tuple $E_i$ that contains $e$ but also potentially affect other tuples in the batch $\mathcal{B}$. The impact on *multiple* terms in the sum of gradients in Eq. (2) prohibits us from properly controlling the sensitivity of $\mathbf{g}(\mathcal{B})$ by clipping each individual gradient $\mathbf{g}(E_i)$. Specifically, *In-batch Negative* is commonly used to obtain negative relations for training large models, due to its simplicity and memory efficiency (for the privacy concerns of other negative sampling methods, see Appx. A.2) (Chen et al., 2020; You et al., 2020; Gao et al., 2021; Radford et al., 2021). Specifically, given a positive relation $e_i^+$, *In-batch Negative* implicitly samples negatives by pairing one end of $e_i^+$ with any entity in other positive relations sampled in the same batch as negative relations (see the upper right of Fig. 1). This practice can lead to the worst case with high

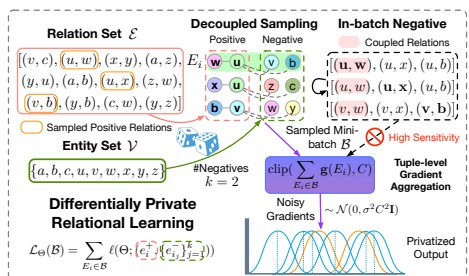

Figure 1: Challenges in differentially private relational learning: Each loss term typically involves coupled relations through negative sampling in a mini-batch, where perturbing one relation may affect multiple loss terms in the same batch (e.g., removing relation $(u, w)$ from the set $\mathcal{E}$ may affect all tuples in $\mathcal{B}$). Decoupled sampling (e.g., pairing negatives from the set $\mathcal{V}$) limits such perturbation to affect at most one relation tuple $E_i$ in a mini-batch.

sensitivity that perturbing one positive relation impacts the whole mini-batch: If a positive relation $e_i^+$ sampled in the batch $\mathcal{B}$ is removed, the loss of every other tuple $E_j \in \mathcal{B}$ will be affected as the entities in $e_i^+$ may be used to form negative relations in $E_j$.

### 3.2 Privacy-preserving Relational Learning

To address the above challenge, we propose to decouple the sampling of negative relations from the set of positive relations. Specifically, the sampling method to form negative relations should not rely on accessing the relation set $\mathcal{E}$ nor leverage sampled positive relations in the same batch. One direct way is to randomly pair one end of the positive relation $e_i^+$ with $k$ entities $(v_{i_1}, \ldots, v_{i_k})$

sampled uniformly at random from the whole entity set $\mathcal{V}$ as negatives $\{e_{i_j}^-\}_{j=1}^k$ [1] for each tuple $E_i$, illustrated in the upper middle of Fig. 1. Now, removing or adding a positive relation will change *at most one* tuple $E_i$ in a mini-batch $\mathcal{B}$. Therefore, by clipping the norm of the gradient of each tuple $\mathbf{g}(E_i)$, we can bound the sensitivity of the gradient sum $\mathbf{g}(\mathcal{B})$, independent of the batch size: The $k$-many negative relations $\{e_{i_j}^-\}_{j=1}^k$ also contribute to the gradient computation of $\mathbf{g}(\mathcal{B})$, but in this new strategy, they only depend on the positive relation $e_i^+$ in the same tuple and their effect is bounded through clipping $\mathbf{g}(E_i)$. Overall, this sampling method is compatible with DP-SGD: Each aggregated gradient $\mathbf{g}(E_i)$ in a mini-batch is clipped and noised as

$$\tilde{\mathbf{g}}(\mathcal{B}) = \frac{1}{b} \left[ \sum_{E_i \in \mathcal{B}} \mathrm{Clip}\left(\mathbf{g}(E_i), C\right) + \mathcal{N}(0, \sigma^2 C^2 \mathbf{I}) \right]. \tag{3}$$

With decoupled negative sampling and gradient obfuscation via Eq. a(3), the privacy analysis of standard DP-SGD holds for relational learning with the loss of Eq. (1), since each relation $e \in \mathcal{E}$ influences the gradient sum at most $C$. The full pipeline to achieve $(\epsilon, \delta)$-DP for relational learning is summarized in Alg. 3, Appx. C.

Another challenge in applying DP-SGD is computing the per-sample gradient, which requires correctly tracking the norm of the gradient for each training sample. This is a common bottleneck in private learning and is amplified in relational learning tasks. Modern privacy libraries such as Opacus (Yousefpour et al., 2021) support tracking the parameter gradient through a training sample when one sample takes only one data point. In the case of relational learning, a relation tuple contains multiple entities, which makes these libraries not directly applicable. A naïve implementation is to hook the parameter gradient through each entity $\mathbf{g}(u|e', E_i) = \frac{\partial \ell(\Theta; E_i)}{\partial \mathbf{z}_{e'}} \cdot \frac{\partial \mathbf{z}_{e'}}{\partial \mathbf{h}_u} \cdot \frac{\partial \mathbf{h}_u}{\partial \Theta}$ during a backward pass. This means that the gradient $\mathbf{g}(E_i)$ of model parameters through one tuple needs to be calculated via all entities in this tuple, i.e., $\mathbf{g}(E_i) = \sum_{e' \in E_i} \sum_{u \in e'} \mathbf{g}(u|e', E_i)$. Computing and caching gradients through each entity $\mathbf{g}(u|e', E_i)$ incurs significant overhead for tuples with large sizes $k$. This issue becomes more pressing for training large models. Next, we aim to address this computational problem.

## 3.3 Efficient Gradient Clipping in Relational Learning

The main computational bottleneck in applying DP-SGD is the tuple size $k$ introduced by relational learning and the lack of support for tracking tuple-level gradients $\mathbf{g}(E)$. The situation worsens when the relational data contain textual attributes and the target model is a language model (Jin et al., 2023a). Specifically, when DP-SGD is applied to sequence models (e.g., Transformers (Vaswani et al., 2017)) that take in multiple tokens, the parameter gradient through each token prediction will be hooked. This means that when the proposed pipeline is applied to LLMs for privacy-preserving relational learning, the parameter gradient through each token $m$ in entity $u$ is actually hooked, which introduces intractable memory cost. Prior work (Lee & Kifer, 2021; Li et al., 2021) proposed some strategies to address this issue for privately fine-tuning LLMs on standard text data, but we find that these techniques are insufficient for relational learning.

Next, we present a tailored approach for efficiently computing the per-tuple gradient $\mathbf{g}(E)$, which exploits the low-rank characterization of per-sample gradient (Goodfellow, 2015) and the structure of $\mathbf{g}(E)$ in relational learning. We use the linear (or embedding) layer of Transformers for demonstration, while non-sequence models can be treated as the case without the token dimension.

For a linear layer in Transformers, its weight matrix is $\mathbf{W} \in \mathbb{R}^{p \times d}$, where $d, p$ are the input and output dimensions, respectively. For a tuple $E$, let $\mathbf{a} \in \mathbb{R}^{K \times M \times d}$ denote the concatenated input, which contains $K = 2(k+1)$ entities, and each entity is associated with $M$ tokens. Let $\mathbf{s} \in \mathbb{R}^{K \times M \times p}$ be the output, where $\mathbf{s}_{i,j} = \mathbf{W}\mathbf{a}_{i,j}$ corresponds to the $j$-th token of the $i$-th entity in the tuple. Denote the gradient w.r.t. $\mathbf{s}_{i,j}$ as $\mathbf{r}_{i,j} = \frac{\partial \ell(\Theta; E)}{\partial \mathbf{s}_{i,j}}$. Then, the gradient of $\mathbf{W}$ through $\mathbf{s}_{i,j}$ can be represented as $\nabla_{\mathbf{W}|\mathbf{s}_{i,j}} \ell = \frac{\partial \ell(\Theta; E)}{\partial \mathbf{s}_{i,j}} \cdot \frac{\partial \mathbf{s}_{i,j}}{\partial \mathbf{W}} = \mathbf{r}_{i,j} \mathbf{a}_{i,j}^T \in \mathbb{R}^{p \times d}$. To compute the per-tuple gradient w.r.t. $\mathbf{W}$, i.e., $\sum_{i=1}^K \sum_{j=1}^M \nabla_{\mathbf{W}|\mathbf{s}_{i,j}} \ell$, it is costly to first compute $\mathbf{r}_{i,j} \mathbf{a}_{i,j}^T$ for each token

---

[1]Decoupled negative sampling may treat entity pairs with observed relations as negatives with a low probability (similar to in-batch negatives), but no obvious harm to utility is observed in practice.

and then sum them over. Instead, a cheaper way is to record $\mathbf{r} = [\cdots, \mathbf{r}_{i,j}, \cdots] \in \mathbb{R}^{K \times M \times p}$ and $\mathbf{a} = [\cdots, \mathbf{a}_{i,j}, \cdots] \in \mathbb{R}^{K \times M \times d}$, and compute $\mathbf{ra}^\top$ to accomplish the sum. This strategy can reduce the memory cost from $\mathcal{O}(KMpd)$ to $\mathcal{O}(KM(p+d)+pd)$. Our experiments use Llama2-7B (Touvron et al., 2023), where $K \in [10, 34]$ and $M = 32$ while $p = d = 4096$ in attention blocks and $p = 32000, d = 4096$ in the embedding block. So, $pd \gg KM(p+d)$ and thus the overall saving based on the above approach is a factor of $\mathcal{O}(KM)$. In addition, some parameter-efficient fine-tuning techniques such as LoRA (Hu et al., 2022) can be incorporated to further reduce the memory cost to $\mathcal{O}(KM(p+d+2r)+(p+d)r)$, where $r$ is the rank of adjustment $\triangle \mathbf{W}$ for parameter $\mathbf{W}$.

## 4 EXPERIMENTS

**Problem Setting & Datasets.** Our experiment design aims to simulate common scenarios of applying relational learning in sensitive domains, where the graph data used to enhance target models contain personal or proprietary relations that need to be protected, such as in applications of e-commerce (Peng et al., 2024), finance (Wu et al., 2023; Ouyang et al., 2022), and healthcare (Gao et al., 2023). We consider two specific use cases: *Cross-category recommendation* - When launching new product lines, RecSys models often face the problem of lacking historical data for prediction (e.g., co-purchase), which can be alleviated by utilizing user purchase history of complementary categories. Sensitive user behaviors contained in these co-purchase relations should be protected. *Cross-regional model deployment* - Financial institutions operate in multiple regions, and their service models (e.g., fraud detection) are normally trained on transaction data collected from major markets and then deployed to multiple regions after fine-tuning. Such practices must comply with data export and protection regulations.

Two publicly available real-world text-attributed graphs with millions of entities/relationships are selected to simulate the two scenarios above: the e-commerce network from Amazon (AMAZ) (McAuley et al., 2015) and the academic network from Microsoft Academic Graph (MAG) (Sinha et al., 2015). In the AMAZ dataset, each entity is a shopping item, and the relation between them indicates that they were co-purchased by customers. It is split into two domains based on the item category: clothing and sports. In the MAG dataset, each entity is a research paper, and the relation between them reflects one cited the other. It is split into two domains based on the region of the main authors: USA and China. In total, four domain-specific subgraphs (see Table 8, Appx. D) are used for relational learning, where the target model is evaluated on the corresponding test domain, such as trained on co-purchased relations from AMAZ-Cloth and tested on AMAZ-Sports.

We conduct a comprehensive set of empirical studies with the objective of addressing two key research questions: **RQ1** Can the target model learn relational knowledge from the training graph with privacy preservation and generalize to relational learning tasks on new test domains? **RQ2** How does negative sampling size $k$ impact the results in relational learning? What about the choice of other hyperparameters, such as the batch size, learning rate, and privacy hyperparameters $\sigma, C$? Does it follow the same rules of the private non-relational setting (Li et al., 2021)?

**Tasks & Metrics.** Privately fine-tuned models are deployed to new test domains under the settings of zero-shot and 16-shot for relation prediction, and 8-shot for entity classification. We use ranking metrics of top@1 precision (PREC@1) and mean reciprocal rank (MRR) to evaluate each model on in-batch negative samples with a batch size of 256 (same as Jin et al. (2023b)) for relation prediction, while Macro-F1 and Micro-F1 for entity classification.

**Implementation Details.** The pretrained models are fine-tuned under the supervision of relational information with the InfoNCE loss and optimized by DP-Adam through our proposed pipeline (see Alg. 3). The privacy loss is tracked through PRV accounting (Gopi et al., 2021). Following prior work on private fine-tuning of LLMs (Li et al., 2021; Yu et al., 2022), we consider privacy levels $\epsilon \in \{4, 10\}$ and $\delta = \frac{1}{|\mathcal{E}_{\text{train}}|}$ for a training set of size $|\mathcal{E}_{\text{train}}|$. Hyperparameters are tuned under given privacy parameters. See Appx. D for other details.

**Baselines.** To the best of our knowledge, our approach is the first for relational learning with differential privacy. DP-GNNs are excluded from baselines due to their insufficiency in properly preserving privacy in relational learning. To compare with feasible privacy-preserving techniques that satisfy DP for relational data, we apply the standard randomized response (RR) baseline to the re-

| Privacy | Target Model | MAG-USA | | MAG-CHN | | AMAZ-Cloth | | AMAZ-Sports | |
|---|---|---|---|---|---|---|---|---|---|
| | | PREC@1 | MRR | PREC@1 | MRR | PREC@1 | MRR | PREC@1 | MRR |
| base model zero-shot | BERT.base | 4.41 | 9.94 | 6.48 | 12.69 | 14.90 | 22.41 | 8.36 | 14.04 |
| | BERT.large | 2.00 | 5.48 | 2.71 | 6.39 | 5.72 | 10.11 | 3.78 | 7.37 |
| | SciBERT | 8.70 | 17.12 | 13.89 | 23.96 | - | - | - | - |
| | LinkBERT.large | 1.09 | 4.01 | 1.46 | 4.75 | 4.01 | 8.60 | 2.06 | 5.37 |
| | Llama2-7B | 4.24 | 8.68 | 5.21 | 9.71 | 19.45 | 27.41 | 6.13 | 10.11 |
| $\epsilon = \infty$ | BERT.base | 28.07 | 39.11 | 41.93 | 53.91 | 36.13 | 47.07 | 29.84 | 39.61 |
| | BERT.large | 26.37 | 37.73 | 40.90 | 53.16 | 36.89 | 47.50 | 29.30 | 39.76 |
| | Llama2-7B | 32.80 | 46.67 | 45.65 | 58.59 | 41.01 | 52.39 | 29.21 | 41.44 |
| $\epsilon = 10$ (RR) | BERT.base | 3.28 | 8.70 | 5.10 | 11.47 | 19.97 | 29.76 | 8.03 | 13.73 |
| | BERT.large | 5.67 | 11.75 | 8.65 | 15.43 | 22.81 | 32.31 | 7.36 | 12.15 |
| | Llama2-7B | 13.64 | 22.33 | 9.92 | 16.67 | 30.39 | 41.48 | 19.63 | 27.66 |
| $\epsilon = 10$ (Ours) | BERT.base | 23.29 | 33.98 | 35.64 | 47.74 | 32.63 | 43.17 | 26.66 | 36.76 |
| | BERT.large | 22.71 | 33.76 | 35.18 | 47.03 | 31.20 | 41.28 | 28.18 | 38.68 |
| | Llama2-7B | 24.07 | 37.53 | 34.58 | 48.76 | 40.16 | 51.25 | 29.54 | 39.90 |
| $\epsilon = 4$ (Ours) | BERT.base | 22.08 | 32.69 | 31.42 | 43.54 | 33.24 | 43.67 | 26.82 | 36.80 |
| | BERT.large | 21.78 | 32.60 | 34.84 | 46.62 | 29.73 | 39.63 | 27.63 | 38.06 |
| | Llama2-7B | 22.55 | 35.47 | 32.50 | 46.68 | 39.67 | 51.09 | 29.25 | 39.35 |

Table 1: Results on **zero-shot** relation prediction with privacy-preserving relational learning.

lation set $\mathcal{E}$ and then perform model fine-tuning on the processed relation set that achieves $\epsilon$-DP (Epasto et al., 2022). Given an entity $u$, for each pair $(u, v), v \in \mathcal{V}, v \neq u$, we apply the randomized response mechanism (Dwork et al., 2014). With probability $p = 1/(1 + \exp(\epsilon))$, the relation label of $(u, v)$ is flipped; otherwise, the original label is kept. Note that this baseline requires $\Theta(N^2)$ time complexity and drastically increases the number of relations for small values of $\epsilon$, which severely limits its applicability.

## 4.1 Evaluation of Privately Fine-tuned Models (RQ1)

In this section, we study the effectiveness of the proposed pipeline by evaluating privately fine-tuned models on new test domains for relation prediction and entity classification. The scale of privacy noise $\sigma$ and the exact privacy loss $\epsilon$ on relational data used for training each model are reported in Table 9, Appx. E. The privacy risks of fine-tuned models are empirically examined via membership inference attacks.

**Relation Prediction.** This task aims to estimate the likelihood of forming a relationship between two entities with specific semantics. Under the *zero-shot* setting, all pretrained language models are privately fine-tuned on relations from the training graph and then are directly deployed on the test domain for inference. This is often faced in cold-start recommendation problems, where the test domain lacks relational information. Results of zero-shot relation prediction in Table 1 show that using co-purchase/citation relations from training graphs to fine-tune language models through our approach can improve their base models' performance on new test domains under DP guarantee $\epsilon = \{4, 10\}$. There is only a modest performance drop compared to the non-private fine-tuned baselines ($\epsilon = \infty$, oracle), which is much smaller than all results from training on relational sets processed by the randomized response mechanism (not computationally feasible for $\epsilon = 4$). This observation validates the effectiveness of privacy-preserving relational learning in light of capturing generalizable relational patterns and knowledge without compromising individual privacy. Among different types of target models, decoder-only models tend to perform worse than encoder models in embedding text (Li & Li, 2024; BehnamGhader et al., 2024), as reflected in the performance disparity between their base models in Table 1. Through (private) relational learning, Llama2-7B can generate rich contextual representations to predict relations and outperform the widely used BERT-based encoder. Next, we consider the *few-shot* setting used for cases like cross-regional model deployment, which is often limited by resource or data availability. The model obtained above is further fine-tuned using 16 training and 16 validation relations from the test domain. Table 2 shows that privately fine-tuned language models still outperform their base models if few-shot fine-tuning is allowed. In particular, they outperform SciBERT (Beltagy et al., 2019) and LinkBERT (Yasunaga et al., 2022a) on the MAG dataset, both of which are pretrained on documents and associated relationships in the scientific domain.

| Privacy | Model | MAG-USA | | MAG-CHN | | AMAZ-Cloth | | AMAZ-Sports | |
| | | PREC@1 | MRR | PREC@1 | MRR | PREC@1 | MRR | PREC@1 | MRR |
|---|---|---|---|---|---|---|---|---|---|
| base model few-shot | BERT.base | 10.24 | 18.94 | 17.10 | 27.84 | 20.42 | 29.74 | 14.70 | 23.46 |
| | BERT.large | 6.57 | 13.88 | 9.61 | 17.75 | 19.57 | 28.69 | 11.23 | 17.80 |
| | SciBERT | 22.27 | 34.24 | 32.42 | 46.10 | - | - | - | - |
| | LinkBERT.large | 21.76 | 31.93 | 35.09 | 47.80 | 13.41 | 19.24 | 23.21 | 30.95 |
| | Llama2-7B | 6.21 | 12.26 | 6.29 | 11.51 | 20.25 | 28.42 | 7.17 | 11.79 |
| $\epsilon = \infty$ | BERT.base | 27.28 | 38.61 | 39.15 | 51.28 | 33.45 | 44.42 | 29.57 | 39.71 |
| | BERT.large | 26.19 | 37.69 | 37.91 | 49.93 | 34.60 | 45.48 | 29.85 | 40.79 |
| | Llama2-7B | 35.45 | 49.30 | 45.89 | 58.84 | 41.42 | 52.59 | 31.92 | 44.83 |
| $\epsilon = 4$ (Ours) | BERT.base | 24.56 | 35.55 | 33.62 | 45.72 | 33.40 | 44.23 | 28.64 | 38.34 |
| | BERT.large | 23.09 | 34.21 | 37.23 | 48.65 | 30.39 | 40.78 | 27.80 | 37.87 |
| | Llama2-7B | 22.88 | 35.94 | 32.07 | 46.22 | 39.94 | 51.10 | 29.78 | 40.27 |

Table 2: Results on **16-shot** relation prediction with privacy-preserving relational learning.

| Privacy | Model | MAG-USA | | MAG-CHN | | AMAZ-Cloth | | AMAZ-Sports | |
| | | Macro-F1 | Micro-F1 | Macro-F1 | Micro-F1 | Macro-F1 | Micro-F1 | Macro-F1 | Micro-F1 |
|---|---|---|---|---|---|---|---|---|---|
| base model few-shot | BERT.base | 2.40 | 3.06 | 2.08 | 3.18 | 9.75 | 16.31 | 7.26 | 8.39 |
| | BERT.large | 2.89 | 4.97 | 2.83 | 3.44 | 4.44 | 15.32 | 1.07 | 2.28 |
| | SciBERT | 4.70 | 10.01 | 5.14 | 6.51 | - | - | - | - |
| | LinkBERT | 0.81 | 1.32 | 1.45 | 1.77 | 10.45 | 36.06 | 0.16 | 10.90 |
| | Llama2-7B | 9.3 | 11.43 | 8.76 | 8.64 | 38.41 | 60.01 | 32.26 | 49.14 |
| $\epsilon = \infty$ | BERT.base | 2.02 | 2.88 | 1.88 | 2.23 | 29.05 | 31.37 | 17.50 | 19.81 |
| | BERT.large | 6.88 | 11.57 | 4.90 | 5.32 | 26.31 | 35.59 | 23.53 | 24.42 |
| | Llama2-7B | 14.97 | 18.77 | 11.52 | 10.85 | 32.94 | 50.65 | 57.53 | 63.15 |
| $\epsilon = 4$ (Ours) | BERT.base | 3.61 | 8.49 | 2.40 | 4.74 | 23.42 | 26.43 | 17.87 | 18.63 |
| | BERT.large | 6.31 | 11.16 | 3.07 | 6.45 | 16.77 | 22.98 | 21.71 | 22.67 |
| | Llama2-7B | 16.55 | 18.59 | 13.56 | 13.29 | 35.43 | 54.85 | 44.74 | 50.47 |

Table 3: Results on **8-shot** entity classification with privacy-preserving relational learning.

**Entity Classification.** This task aims to investigate whether injecting relational information helps language models classify entities with textual attributes in adjacent domains. It is motivated by the above observation, where introducing structural knowledge between entities can go beyond contextual semantics and help models refine internal entity representations across domains. We use the language model as an encoder and attach a classifier to take entity embeddings for classification. The parameters of language models are frozen, where only limited examples are used to initialize the classifier. The entity classes are coarse-grained category names from AMAZ and MAG networks. 8 labeled training and 8 validation entities of each class are used for training, while thousands of new entities are reserved for testing. Table 3 shows that models fine-tuned privately on relational data generally produce entity embeddings of better quality than those directly generated from base models. In some cases, private models outperform non-private models ($\epsilon = \infty$), which can be attributed to the regularization effect of DP-SGD under the setting of limited examples. The performance drop on AMAZ-cloth is due to the potential misalignment between the objective of relation-based fine-tuning and entity classification, which has been observed in the non-private relational learning setting by Xie et al. (2023) and in the private non-relational setting by Li et al. (2021).

**Privacy Attacks.** We perform the membership inference attacks (MIAs) on relation samples to empirically estimate the privacy risk of models fine-tuned on the training graph. The high computational cost of training multiple copies of shadow models (Shokri et al., 2017) makes it intractable to perform such attacks on LLM-based relational learning. Thus, we consider an unsupervised approach to determine the membership of target relations

| Method | $\epsilon$ | MAG-USA | | MAG-CHN | |
| | | TPR@0.05 FPR | WSRT $p$-value | TPR@0.05 FPR | WSRT $p$-value |
|---|---|---|---|---|---|
| Non-DP | $\infty$ | 0.0687 | 3.14e-82 | 0.0551 | 1.39e-48 |
| Ours | 10 | 0.0672 | 4.36e-54 | 0.0480 | 4.72e-25 |
| Ours | 4 | 0.0600 | 1.51e-46 | 0.0469 | 6.69e-20 |

Table 4: Attack results against LLama2-7B fine-tuned on training graphs of MAG-USA/CHN. A smaller $p$-value in the Wilcoxon test shows the difference observed between member and non-member distributions is statistically significant, indicating more leakage.

based on a distance-based score function (He et al., 2021; Wang & Wang, 2023). We follow a similar setting as He et al. (2021), where the adversary has knowledge of the target dataset's entity attributes. The attack relies on the posteriors of entity embeddings obtained from the target model to

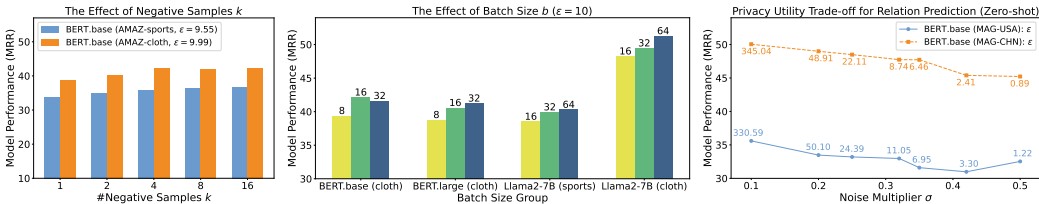

Figure 2: Effects of negative sample $k$, batch size $b$, and noise multiplier $\sigma$ for zero-shot relation prediction with privacy preservation.

measure the score between entity pairs and classify the membership of their relations. To evaluate the attack, we use the True-Positive Rate (TPR) at low False-Positive Rates (FPR) (Carlini et al., 2022), and $p$-value of the Wilcoxon signed-rank test (WSRT) (Wilcoxon, 1992) conducted on the score distributions obtained over relations contained in and not in training (Kim et al., 2024). The small $p$-value suggests that the observed difference between two distributions is statistically significant. From Table 4.1, we observe that attacks on non-private fine-tuned models ($\epsilon = \infty$) have a higher success rate than models trained by our approach on both datasets, where the stricter privacy budget of $\epsilon$ leads to lower success rate of MIAs both in terms of the TPR at 0.05 FPR and the statistical test. We also include the plotting of score distributions obtained from different target models in Fig. 3, Appx. E.

## 4.2 UTILITY, PRIVACY AND COMPUTATIONAL EFFICIENCY TRADE-OFFS (RQ2)

In this section, we study the trade-offs between utility, privacy, and computational complexity in privacy-preserving relational learning. We first investigate the hyperparameters of negative sampling $k$, batch size $b$, and learning rate $\eta$ in a realistic setting, where the training steps are fixed. Fig. 2 (Left) shows the impact of negative sampling in relational learning: Increasing $k$ generally improves prediction performance while with a rapidly decreasing marginal benefit. To achieve a trade-off between performance and computational complexity, the optimal region is located at $k \in [4, 8]$. Fig. 2 (Middle) shows the effect of batch size $b$ on different models under the same privacy budget: Larger $b$ leads to better model performance and quick convergence, especially for Llama2-7B. This observation is consistent with non-relational private learning, where increasing $b$ achieves a better signal-to-noise ratio between the sum of clipped gradients and the Gaussian noise added via Eq. (3). The joint effect of batch size $b$ and learning rate $\eta$ is further studied and depicted in Fig. 4 (Left), Appx. F: Larger batches and learning rates together lead to good performance under fixed training steps, which echoes the findings in privately fine-tuning LLMs on standard text data (Li et al., 2021). The main obstacle to using larger $b$ is the linearly increased privacy computing and memory cost associated with controlling per-sample gradients as discussed in Sec. 3.3.

Next, we study how privacy parameters impact model utility. Fig. 2 (Right) plots the privacy-utility curve of BERT.base on zero-shot relation prediction over MAG-USA/CHN datasets using different privacy budgets $\epsilon$ by adjusting noise multiplier $\sigma$ while keeping other parameters constant. In this case, the scale of privacy noise added solely determines the privacy leakage, where the model performance decays proportionally as the value of $\sigma$ increases. The threshold of norm clipping $C$ does not affect the privacy budget $\epsilon$ but is crucial to the utility performance of DP models (Bu et al., 2024), and its impact on relational learning tasks is shown in Fig. 4 (Right), Appx. åF. Picking a threshold $C$ that is larger than the actual gradient norm means that most clipping through Eq. (3) is not effective, and the noise $\sigma C$ is added more than necessary. In general, small values of $C$ work better for relational learning, which aligns with the general practice and observation of DP learning on non-relational data in both vision and language tasks (Tramer & Boneh, 2021; Li et al., 2021).

## 5 RELATED WORK

**LLMs with Relational Data.** Extensive work has focused on using relational data to enhance foundation models, especially for fine-tuning LLMs on graphs, in light of their strong generalization ability. These methods can be classified into two types. *Objective-only:* Yasunaga et al. (2022b); Duan et al. (2023); Xie et al. (2023) proposed to associate entity representations from LLMs with

relational information by optimizing the objective based on specific graph tasks. For example, relation prediction is a typical task in unsupervised graph learning, as adopted in this work. *Graph-encoder-based:* Chien et al. (2022); Yasunaga et al. (2022a); Zhu et al. (2024); Xie et al. (2023); Jin et al. (2023b) pair LLMs with a graph encoder (e.g., GNNs (Kipf & Welling, 2017)) to incorporate relational information in an end-to-end manner, where LLMs act as feature extractors for textual attributes, and their output with associated graph topology is fed into GNNs for aggregation and prediction. These models may be privatized by combining the approach proposed in this work with the privatized method for graph encoders (Sajadmanesh et al., 2023; Chien et al., 2024), though the entire pipeline could be complex and beyond the scope of this work.

**Privacy-preserving for LLMs.** Data privacy in LLMs focuses on safeguarding sensitive information that could be exposed during operations (Yao et al., 2024). Recent efforts have utilized DP-SGD for both pretraining and fine-tuning LLMs. For instance, Anil et al. (2022) trained a privacy-preserving BERT.large model from scratch. However, due to the resource-intensive nature of LLMs, the focus has shifted towards private fine-tuning of publicly pretrained models. Hoory et al. (2021) explored private full fine-tuning of BERT models with domain-specific data, while recent advancements in this field include the works of Basu et al. (2021); Kerrigan et al. (2020); Senge et al. (2022); Li et al. (2021). There is growing interest in efficient private fine-tuning techniques. Yu et al. (2022) applied parameter-efficient fine-tuning methods for private fine-tuning of LLMs, and Li et al. (2021) introduced ghost clipping to accelerate gradient clipping in DP-SGD. However, these methods primarily address privacy concerns for standard text data. In contrast, our work extends these privacy-preserving techniques to relational data, filling an important gap in this research area.

**Privacy-preserving Graph Learning.** Significant research has focused on privacy-preserving graph embedding and learning algorithms with DP guarantees (Li et al., 2023b). Daigavane et al. (2021) proposed a privacy-preserving approach for training GNNs via extensions of DP-SGD. Olatunji et al. (2023) adopted teacher-student models to enable the DP release of GNNs. Sajadmanesh et al. (2023) improved utility-privacy trade-offs by decoupling feature propagation and model training, and their work further got extended in subsequent studies (Sajadmanesh & Gatica-Perez, 2024; Chien et al., 2024). These methods specialize in generating private node representations and privatizing the graph exposed to encoders for feature propagation and aggregation, which do not mitigate privacy risks when relational information is used for supervision.

## 6 CONCLUSION

Leveraging relational data to enhance AI models holds great promise. This work proposes a novel privacy-preserving training pipeline that addresses the unique privacy and computational challenges in relational learning by decoupling the dependencies in sampled relations for training and exploiting gradient structure through individual samples for efficient clipping. We consider scenarios frequently encountered in applying relational learning to fine-tune pretrained language models and enforce privacy guarantees on the relationships used for training. Our study on privacy-preserving relational learning shows that fine-tuning language models with our approach can significantly improve their performance on new test domains while keeping the relational training data differentially private. We further explore the privacy, utility, and computational efficiency trade-offs and conduct an extensive study on hyperparameter selection for relational learning in private settings.

ACKNOWLEDGMENTS

This work is supported by NSF awards CCF-2402816, IIS-2239565, and 2023 JPMC faculty award.

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

## A  PRIVACY RISKS IN RELATIONAL LEARNING

Relational learning aims to inject observed relational information between entities into target models, commonly captured by graph data $\mathcal{G} = (\mathcal{V}, \mathcal{E}, X)$. Suppose the representation of each entity $u \in \mathcal{V}$ is obtained from a model parameterized by $\Theta$ encoding its attribute, i.e., $\mathbf{h}_u = f_\Theta(X_u)$. To make the target model relation-aware, the refinement is usually through optimizing the parameter $\Theta$ via a relation-based loss, generally with the form as $\ell(\Theta; (e_i^+, \{e_{i_j}^-\}_{j=1}^k))$. The following uses pairwise relation as an example: $e_i^+ = (u, v)$ is observed in a relation set $\mathcal{E}$, usually termed as positive relation, while the unobserved relation $e_{i_j}^- = (u, w) \notin \mathcal{E}$ is referred as negative. Let $\Gamma(\cdot)$ be a score function that measures or labels the relationship between entities based on their embeddings output by the model. The intuition behind common loss $\ell$ used in relational learning (e.g., InfoNCE loss (Oord et al., 2018) and pairwise Hinge loss (Bordes et al., 2013)), is to promote positive relations $\mathbf{z}_{e_i^+} = \Gamma(\mathbf{h}_u, \mathbf{h}_v)$ having a higher score than paired negative relations $\mathbf{z}_{e_{i_j}^-} = \Gamma(\mathbf{h}_u, \mathbf{h}_w)$. The overall objective can be written as

$$\Theta^* = \arg\min \sum_{e_i^+ \in \mathcal{E}} \ell(\Theta; (\mathbf{z}_{e_i^+}, \{\mathbf{z}_{e_{i_j}^-}\}_{j=1}^k)). \tag{4}$$

Here, $\Theta$ is the parameter of any target model that can encode entity attributes. The relational information is only exposed to the model through the loss function $\ell$ as *labels for supervision* and carried into model parameters during backpropagation. The privacy risk of relational learning through the above procedure can be mitigated by carefully limiting and obfuscating the update of model parameters induced by individual relations, which is the main objective studied in this work.

## A.1 DIFFERENTIALLY PRIVATE GNNS ARE INSUFFICIENT FOR RELATIONAL LEARNING WITH DIFFERENTIAL PRIVACY

Graph neural networks (GNNs) (Kipf & Welling, 2017) are one of the most popular encoders that can be directly applied to relational data for obtaining entity embeddings. The key mechanism of GNNs is message-passing, where information is propagated and aggregated among neighborhoods of entities along graph topology. A typical GNN consists of $L$ sequential graph convolution layers, which is formulated as

$$\mathbf{h}_u^{(l)} = \text{upd}\left(\text{agg}\left(\{\mathbf{h}_v^{(l-1)} : \forall v \in \mathcal{N}(v)\}\right); \Theta_g^{(l)}\right), \tag{5}$$

where $\mathcal{N}(u) = \{v : (u,v) \in \mathcal{E}\}$ denotes the set of neighboring entities to entity $u$, and $\mathbf{h}_v^{(l-1)}$ is the embedding of an neighboring entity $v$ at layer $l-1$. $\text{agg}(\cdot)$ is a differentiable, permutation invariant aggregation function (e.g., sum, mean, or max). $\text{upd}(\cdot)$ is a learnable function, such as a multi-layer perception (MLP), parameterized by $\Theta_g^{(l)}$ that takes the aggregated embeddings and outputs the updated embedding for the root entity $\mathbf{h}_v^{(l)}$.

Next, we show why previous methods that build differentially private GNNs cannot handle the challenges of relational learning with differential privacy. DP-GNNs (Daigavane et al., 2021; Zhang et al., 2024b; Sajadmanesh et al., 2023; Sajadmanesh & Gatica-Perez, 2024; Chien et al., 2024) primarily address the privacy issue during graph data encoding. Specifically, in GNN models, perturbing a node or an edge affects not only itself and its direct neighbors but also multi-hop neighbors through recursive layer-wise message passing, as shown in Eq. (5). Existing efforts mainly focus on limiting such correlation in message passing (e.g., node degree and number of hops) so that the sensitivity of graph encoding in GNNs can be bounded, which makes it feasible for DP training on entity- or graph-level tasks. Unfortunately, these techniques are insufficient for relational learning because coupled relations in the loss (e.g., Eq. (4)) make the gradient not decomposable into specific privacy-preserved samples and thus still leaves privacy-preserving relational learning an open problem. We aim to solve this problem in this work.

## A.2 PRIVACY CONCERNS WITH OTHER TYPES OF NEGATIVE SAMPLING IN RELATIONAL LEARNING

As discussed in Sec. 3.1, the main challenge in applying DP-SGD to privacy-preserving relational learning is the coupled sampling of positive and negative relations during training. Here, we showcase the coupling caused by other types of negative sampling. *Random Negative Sampling* is one of the most widely used methods (Yang et al., 2024). Given a positive relation $e_i^+ = (u,w)$, it uniformly samples negative relations containing either entity $u$ or $w$ from the complement set $\bar{\mathcal{E}} = \binom{\mathcal{V}}{2} \setminus \mathcal{E}$, for example, $e_{i_j}^- = (u,v) \in \bar{\mathcal{E}}$. This method requires access to $\mathcal{E}$ to obtain $\bar{\mathcal{E}}$ for negative selection and makes the sampled negative relations dependent on the positive relations that share common entities. If an originally negative relation $(u,v)$ is flipped as positive and added to the relation set $\mathcal{E}$, all tuples in the batch $\mathcal{B}$ that previously sampled $(u,v)$ as negative relations will be affected. In the worst-case scenario, it may affect the entire mini-batch, introducing a high sensitivity that cannot be properly controlled by clipping the gradient of each tuple. This observation also makes all sampling methods that require accessing the relation set $\mathcal{E}$ ineligible for relational learning in private settings, such as *Popularity-based Negative Sampling* and the family of *Hard Negative Sampling* that generates true negative samples.

## B OTHER RELATED WORK

**Private Graph Embedding Methods.** Graph embedding encodes nodes into low-dimensional vectors, preserving topological information (Hamilton et al., 2017). Xu et al. (2018) proposed a private

---

**Algorithm 1:** DP-SGD from Abadi et al. (2016)

---

**Input:** Training data $x_1, \ldots, x_N$, loss function $\mathcal{L}(\Theta) = \frac{1}{N}\sum_i \mathcal{L}(\Theta, x_i)$; Parameters: learning rate $\eta_t$, batch size $b$, gradient norm threshold $C$, noise multiplier $\sigma$ or privacy budget $\epsilon$.
**Initialize** find the optimal value of $\sigma$ via calibration if $\epsilon$ is given.
**for** $t = 1$ **to** $T$ **do**
   **Subsampling**
Randomly sample $\mathcal{B}_t$ with sampling probability $b/N$
   **Compute Gradient**
For each $x_i \in \mathcal{B}_t$, compute $\mathbf{g}_t(x_i) \leftarrow \nabla_{\Theta_t}\mathcal{L}(\Theta_t, x_i)$
   **Gradient Clipping**
$\bar{\mathbf{g}}_t(x_i) \leftarrow \mathbf{g}_t(x_i) / \left[\max\left(1, \frac{||\mathbf{g}_t(x_i)||_2}{C}\right)\right]$
   **Add Noise**
$\tilde{\mathbf{g}}_t \leftarrow \frac{1}{b}\left[\sum_i \bar{\mathbf{g}}_t(x_i) + \mathcal{N}(0, \sigma^2 C^2 \mathbf{I})\right]$
   **Parameter Update**
$\Theta_{t+1} \leftarrow \Theta_t - \eta_t \tilde{\mathbf{g}}_t$
**end for**
**Output** $\Theta_T$ and calculate the overall privacy cost $(\epsilon, \delta)$ using an accounting method if $\sigma$ is given.

---

network embedding method using objective perturbation in DeepWalk (Perozzi et al., 2014) but faced scalability issues for complex sensitivity calculations. Zhang & Ni (2019) addressed these issues by applying a Lipschitz condition (Raskhodnikova & Smith, 2016) and gradient clipping. Epasto et al. (2022); Wei et al. (2024) studied DP PageRank methods, which can be leveraged to generate DP graph embedding as well. These methods specialize in preventing privacy leakage during generating node embeddings but cannot protect the privacy of relations that are used for supervision studied in this work.

**Private Aggregation of Teacher Ensembles.** PATE and its variants (Papernot et al., 2018) are an alternative privacy-preserving method to achieve DP in machine learning, which leverages an ensemble of teacher models that are trained on disjoint datasets containing sensitive data. These models are not published but instead used as teacher models for a separate student model. The student model cannot access any single teacher model or the underlying data. It learns to predict an output chosen by noisy voting performed across all teacher models. The student is trained on a publicly available, unlabelled dataset, where the labels come from the aggregate votes of the teachers. The availability of public data is one notable limitation of PATE, as well as the complexity of training a collection of teacher models. Olatunji et al. (2023) proposed a PATE framework to release GNNs with node-DP for node-level tasks. Adopting PATE in graph settings faces challenges of low utility due to the physical separation of datasets in graph learning that destroys structural information and limited generality beyond graph classification settings.

**Contrastive Learning with Differential Privacy.** Existing studies on private contrastive learning aim to eliminate the risk of sample correlation in contrastive losses and thus protect the privacy of individual training samples. Li et al. (2022) proposed to add privacy noise to the similarity matrix between pairs of inputs to reduce the sensitivity of gradients w.r.t. the contrastive loss. Kong et al. (2025) extended it to similarity-based loss functions by bounding the pairwise similarity gradients. Bao et al. (2024) proposed to train vision models with the mixup technique under DP by leveraging augmentation multiplicity. These methods focus on learning representations of non-relational samples differentially private by contrastive views, but they cannot be used to address the privacy challenges of coupled relations in training models on relational data.

## C  LEARNING PIPELINES OF STANDARD DP AND RELATIONAL SETTINGS

DP-SGD (see Alg. 1) (Song et al., 2016; Abadi et al., 2016) is proposed for training deep learning models on (non-relational) samples with a privacy guarantee. DP-Adam (see Alg. 2) works similarly as regular Adam (Kingma & Ba, 2015) but performs updates and moment accumulation with privatized gradients. The gradient privatization part is the same as that performed in DP-SGD, where the privacy analysis and guarantees for DP-SGD still hold for DP-Adam due to the post-processing

---

**Algorithm 2:** DP-Adam (Kingma & Ba, 2015; Abadi et al., 2016)

---

**Input:** Training data $x_1, \ldots, x_N$, loss function $\mathcal{L}(\Theta) = \frac{1}{N} \sum_i \mathcal{L}(\Theta, x_i)$; Parameters: learning rate $\eta_t$, batch size $b$, gradient norm threshold $C$, noise multiplier $\sigma$ or privacy budget $\epsilon$, initial moment estimates $m_0, v_0$, exponential decay rates $\beta_1, \beta_2$, avoid division-by-zero constant $\gamma$.

**Initialize** find the optimal value of $\sigma$ via calibration if $\epsilon$ is given.

**for** $t = 1$ **to** $T$ **do**

    **Subsampling**

Randomly sample $\mathcal{B}_t$ with sampling probability $b/N$

    **Compute Gradient**

For each $x_i \in \mathcal{B}_t$, compute $\mathbf{g}_t(x_i) \leftarrow \nabla_{\Theta_t} \mathcal{L}(\Theta_t, x_i)$

    **Gradient Clipping**

$\bar{\mathbf{g}}_t(x_i) \leftarrow \mathbf{g}_t(x_i) / \left[ \max\left(1, \frac{\|\mathbf{g}_t(x_i)\|_2}{C}\right) \right]$

    **Add Noise**

$\tilde{\mathbf{g}}_t \leftarrow \frac{1}{b} \left[ \sum_i \bar{\mathbf{g}}_t(x_i) + \mathcal{N}(0, \sigma^2 C^2 \mathbf{I}) \right]$

    **Parameter AdamUpdate**

$m_{t+1} \leftarrow \beta_1 \cdot m_t + (1 - \beta_1) \cdot \tilde{\mathbf{g}}_t, \; v_{t+1} \leftarrow \beta_2 \cdot v_t + (1 - \beta_2) \cdot \tilde{\mathbf{g}}_t^2$

$\hat{m}_{t+1} \leftarrow m_{t+1}/(1 - \beta_1^t), \hat{v}_{t+1} \leftarrow v_{t+1}/(1 - \beta_2^t)$

$\Theta_{t+1} \leftarrow \Theta_t - \eta_t \cdot \hat{m}_{t+1}/\left(\sqrt{\hat{v}_{t+1}} + \gamma\right)$

**end for**

**Output** $\Theta_T$ and calculate the overall privacy cost $(\epsilon, \delta)$ using an accounting method if $\sigma$ is given.

---

**Algorithm 3:** Learning on Relational Data with Differential Privacy

---

**Input:** target model $f_\Theta$ (e.g., LLMs), graph $\mathcal{G} = (\mathcal{V}, \mathcal{E}, X)$, scoring function $\Gamma$, loss function $\ell$; Parameters: learning rate $\eta_t$, batch size $b$, number of negative samples $k$, threshold of gradient norm $C$, noise multiplier $\sigma$ or privacy budget $\epsilon$.

**Initialize** find the optimal value of $\sigma$ via calibration if $\epsilon$ is given.

**for** $t = 1$ **to** $T$ **do**

    **Subsampling**

I. Randomly sample $\mathcal{B}_t$ from $\mathcal{E}$ with sampling ratio $b/|\mathcal{E}|$.

II. For each sampled positive relation $e_i^+$ in the batch, randomly sample $k$ entities $(v_{i_1}, \ldots, v_{i_k})$ without replacement from $\mathcal{V}$ and pair them with one end of $e_i^+$ as negatives $\{e_{i_j}^-\}_{j=1}^k$, which forms a tuple of $k+1$ relations as $E_i = (e_i^+, \{e_{i_j}^-\}_{j=1}^k)$.

    **Compute & Aggregate Gradient**

$\mathbf{g}_t(E_i) = \sum_{e' \in E_i} \sum_{u \in e'} \frac{\partial \ell(\Theta; E_i)}{\partial \mathbf{z}_{e'}} \cdot \frac{\partial \mathbf{z}_{e'}}{\partial \mathbf{h}_u} \cdot \frac{\partial \mathbf{h}_u}{\partial \Theta}$, where $\mathbf{z}_{e'} = \Gamma(\mathbf{h}_u, \mathbf{h}_v)$ for relation $e' = (u, v)$ and $\mathbf{h}_u = f_\Theta(X_u)$ for entity $u$. Note that the actual computation of $\mathbf{g}_t(E_i)$ is performed by the efficient approach proposed in Sec. 3.3.

    **Gradient Clipping & Add Privacy Noise**

$\tilde{\mathbf{g}}_t \leftarrow \frac{1}{b} \sum_{E_i \in \mathcal{B}_t} \left[ \mathbf{g}_t(E_i)/\max\left(1, \|\mathbf{g}_t(E_i)\|_2/C\right) + \mathcal{N}(0, \sigma^2 C^2 \mathbf{I}) \right]$

    **Parameter Update**

$\Theta_{t+1} \leftarrow \Theta_t - \eta_t \tilde{\mathbf{g}}_t$

**end for**

**Output** $\Theta_T$ and calculate the overall privacy cost $(\epsilon, \delta)$ using an accounting method if $\sigma$ is given.

---

property of DP (Dwork et al., 2014). We use DP-Adam (see Alg. 2) as the default optimizer, same as previous work (Li et al., 2021; Yu et al., 2022). The proposed privacy-preserving relational learning pipeline is summarized in Alg. 3.

## D EXPERIMENTAL DETAILS

**Datasets.** Item and paper titles are used as textual attributes associated with entities in the Amazon e-commerce network (AMAZ) (McAuley et al., 2015) and the Microsoft Academic Graph

| AMAZ-Cloth | | AMAZ-Sports | |
|---|---|---|---|
| Label | Name | Label | Name |
| 0 | girls | 0 | accessories |
| 1 | men | 1 | action sports |
| 2 | novelty | 2 | boating & water sports |
| 3 | luggage | 3 | clothing |
| 4 | baby | 4 | cycling |
| 5 | fashion watches | 5 | baby |
| 6 | shoes | 6 | exercise & leisure sports |
| 7 | boys | 7 | fan shop |
| 8 | adidas | 8 | golf |
| | | 9 | hunting & fishing & game room |
| | | 10 | outdoor gear |
| | | 11 | fitness |
| | | 12 | paintball & airsoft |
| | | 13 | racquet sports |
| | | 14 | snow sports |
| | | 15 | team sports |

Table 5: Class names of the AMAZ dataset.

| Target Model | BERT.base | BERT.large | Llama2-7B |
|---|---|---|---|
| DP Guarantee $(\epsilon, \delta)$ | $(-,1/|\mathcal{E}_{\text{train}}|)$ | $(-,1/|\mathcal{E}_{\text{train}}|)$ | $(-,1/|\mathcal{E}_{\text{train}}|)$ |
| Clipping threshold $C$ | 1 | 1 | 1 |
| Noise multiplier $\sigma$ | [0.3, 0.5] | [0.3, 0.5] | [0.3, 0.5] |
| LoRA rank $r$ | {2,4,8,16} | {2,4,8,16} | {2,4,8,16} |
| LoRA alpha $\alpha$ | 16 | 16 | 16 |
| LoRA dropout | [0, 0.2] | [0, 0.2] | [0, 0.2] |
| Target module | query, key, value, dense | query, key, value, dense | q_proj, v_proj |
| Batch size $B$ | {8, 16, 32, 64} | {8, 16, 32, 64} | {12, 16, 32, 64, 128} |
| Learning rate $\eta$ | $[10^{-4}, 10^{-6}]$ | $[10^{-4}, 10^{-6}]$ | $[10^{-4}, 10^{-6}]$ |
| LR scheduler | linear | linear | cosine |
| Weight decay $\lambda$ | $[0, 10^{-3}]$ | $[0, 10^{-3}]$ | 0 |
| Negative sample $k$ | {4, 6, 8, 16} | {4, 6, 8, 16} | {4, 8, 12, 16} |

Table 6: Hyperparameter search range for different models.

| Target Model | License | Model Card |
|---|---|---|
| BERT.base | Apache License 2.0 | `https://huggingface.co/google-bert/bert-base-uncased` |
| BERT.large | Apache License 2.0 | `https://huggingface.co/google-bert/bert-large-uncased` |
| SciBERT | Apache License 2.0 | `https://huggingface.co/allenai/scibert_scivocab_uncased` |
| LinkBERT.large | Apache License 2.0 | `https://huggingface.co/michiyasunaga/LinkBERT-large` |
| Llama2-7B | Meta Community License | `https://huggingface.co/meta-llama/Llama-2-7b-hf` |

Table 7: Model card of pretrained language models.

| Dataset | #Entity | #Relation | #Entity (Test) | #Classes | #Relation (Test) | Test Domain |
|---|---|---|---|---|---|---|
| AMAZ-Cloth | 960,613 | 4,626,125 | 476,510 | 9 | 10,000 | AMAZ-Sports |
| AMAZ-Sports | 357,936 | 2,024,691 | 129,669 | 16 | 10,000 | AMAZ-Cloth |
| MAG-USA | 132,558 | 702,482 | 6,653 | 40 | 63,635 | MAG-CHN |
| MAG-CHN | 101,952 | 285,991 | 6,534 | 40 | 34,603 | MAG-USA |

Table 8: Dataset statistics for evaluation.

(MAG) [2] (Sinha et al., 2015), respectively. OpenAlex API [3] (Priem et al., 2022) is used to obtain metadata of papers in the MAG dataset as the Microsoft Academic service has been retired. For some items/papers, we concatenate their titles with the corresponding description/abstract following Jin et al. (2023b) since the text of the title is too short. The max length of the input token $M$ is set to 32. The semantics of relational information used for supervision are "item-co-purchased-item" and "paper-cited-paper" for AMAZ and MAG networks, respectively. To mimic the case in the cross-category recommendation, two subgraphs are selected from AMAZ that only contain items belonging to the category of clothing (AMAZ-Cloth) and sports (AMAZ-Sports). For entity classification, the class names of the AMAZ dataset are listed in Table 5. Based on the geographic metadata of the main authors, we select two subgraphs from MAG containing papers written by authors from the United States (MAG-USA) and China (MAG-CHN) to simulate the case in cross-regional model deployment. The coarse-grained class of papers in the MAG dataset is refined by selecting the Top-K-occurrence of 349-class obtained from Open Graph Benchmark [4] (Hu et al., 2020) and merging the rest classes into one. The processed data is available at https://zenodo.org/records/15186566.

**Target Models.** Off-the-shelf pretrained language models: BERT (Devlin et al., 2019), a language model pretrained with objectives of masked language modeling (MLM) and next sentence prediction on Wikipedia and BookCorpus, with parameters of 110M (base) and 340M (large). SciBERT (Beltagy et al., 2019) is trained on 1.14M paper abstracts and full text from Semantic Scholar under the same pertaining as BERT. LinkBERT (Yasunaga et al., 2022b) is pretrained with MLM as BERT and the relation-based objective for predicting the linkage between documents. Note that some documents and relations in the MAG dataset may be used during the pretraining of SciBERT and LinkBERT, which potentially causes some data leakage. Llama2-7B (Touvron et al., 2023) is one of the most popular open-source pretrained and fine-tuned LLMs with 7 billion parameters.

**Environment.** We use a server with two AMD EPYC 7543 CPUs, 512GB DRAM, and NVIDIA Quadro RTX 6000 (24GB) GPUs for BERT-based models and A100 (80GB) GPUs for Llama2-7B models. The codebase is built on PyTorch 2.1.2, Transformers 4.23.0, PEFT 0.10.0, and Opacus 1.4.1. The source code is included and should be paired with the Transformers and PEFT packages from HuggingFace and the Opacus library specified above.

**Private Fine-tuning.** We use DP-Adam, a variant from DP-SGD, as the default optimizer for updating model parameters in a privacy-preserving manner: given privacy parameters of noise multiplier $\sigma$ (or calibrated $\sigma$ if $\epsilon$ is given) and clipping threshold of gradient norm $C$, with a learning rate $\eta$ from `1e-4` to `1e-6`, weight decay from `0` to `1e-3`, first 10% as warm-up steps. The batch size of testing is set to 256 for relation prediction, which follows Jin et al. (2023b) that uses in-batch negatives for computing ranking metrics. We use gradient accumulation over multiple mini-batches to simulate training at the expected batch size in case the actual memory needs exceed the physical memory limit of VRAM for large pretrained models. We search optimal training hyperparameters under given privacy parameters, where their ranges are summarized in Table 6. All pretrained model weights are publicly available and directly downloaded from Huggingface under proper licenses listed in Table 7.

**Inference Setting.** Once the model is privately fine-tuned on relational data, it is deployed for inference under two settings for relation prediction and entity classification on the corresponding test domains (see Table 8): *Zero-shot*, where the model is directly used without further training on samples from the test domain. This setting is only applied for relation prediction, where the dot product between entity embeddings is used as the scoring function $\Gamma$ for inference. *Few-shot*, where limited labels from the test domain are provided to further fine-tune the target models obtained after privacy-preserving relational learning. This setting is used for both relation prediction and entity classification (the classifier requires some labels for initializing parameters), corresponding to the scenario of relational data scarcity from the test domain and limited resources to perform full domain-specific fine-tuning.

---

[2]ODC-BY License, refer to https://opendatacommons.org/licenses/by/1-0/

[3]CC0 License, refer to https://creativecommons.org/public-domain/cc0/

[4]ODC-BY License

| Privacy | Model | MAG-USA | | | MAG-CHN | | | AMAZ-Cloth | | | AMAZ-Sports | | |
|---------|-------|---------|-----|-----|---------|-----|-----|------------|-----|-----|-------------|-----|-----|
| | | $\sigma$ | $\epsilon$ | $r$ | $\sigma$ | $\epsilon$ | $r$ | $\sigma$ | $\epsilon$ | $r$ | $\sigma$ | $\epsilon$ | $r$ |
| $\epsilon = 10$ | BERT.base | 0.32 | 9.95 | 4 | 0.32 | 8.74 | 2 | 0.3 | 9.71 | 2 | 0.3 | 9.06 | 8 |
| | BERT.large | 0.34 | 8.72 | 4 | 0.33 | 8.56 | 2 | 0.3 | 9.94 | 8 | 0.32 | 7.69 | 8 |
| | Llama2-7B | 0.378 | 7.91 | 4 | 0.357 | 8.16 | 4 | 0.326 | 8.50 | 8 | 0.315 | 8.83 | 8 |
| $\epsilon = 4$ | BERT.base | 0.42 | 3.30 | 8 | 0.4 | 3.99 | 2 | 0.4 | 3.34 | 2 | 0.4 | 2.65 | 2 |
| | BERT.large | 0.42 | 3.82 | 4 | 0.41 | 3.32 | 2 | 0.376 | 4.00 | 8 | 0.4 | 3.27 | 2 |
| | Llama2-7B | 0.456 | 3.97 | 4 | 0.433 | 4.00 | 4 | 0.4 | 4.00 | 8 | 0.4 | 3.88 | 8 |

Table 9: Privacy loss $\epsilon$ of model fine-tuning on relational data. $\sigma, \epsilon, r$ are noise multiplier, privacy loss, and LoRA rank.

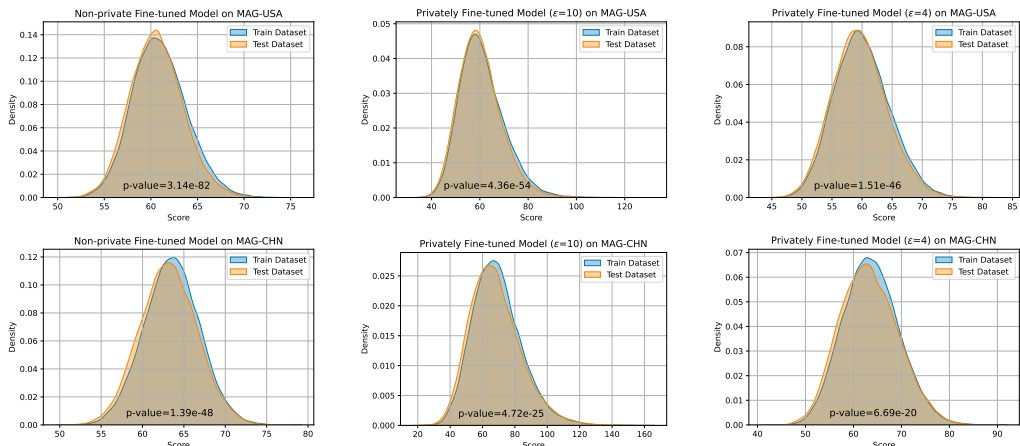

Figure 3: Membership inference attacks on target models fine-tuned non-privately vs privately (ours) over MAG-USA/CHN datasets. The distributions of relationship scores between each entity pair in the training and test sets are plotted. The p-value obtained from the statistical test using the Wilcoxon signed-rank test (Wilcoxon, 1992) is reported, where a small $p$-value suggested that the observed difference between the score distributions of the training and test set is statistically significant.

# E    DETAILS FOR STUDIES IN SECTION 4.1

After privately fine-tuning target models on realtions from the training graph, we use the PRV accounting (Gopi et al., 2021) to track privacy loss and convert it to $(\epsilon, \delta)$-DP. Table 9 summarizes the values of noise multiplier $\sigma$ used and the actual privacy loss $\epsilon$ for training each model on sensitive relational data, which corresponds to the results reported in Table 1 under the zero-shot setting and in Tables 2, 3 under the few-shot setting. Models under the few-shot setting have the same privacy loss as zero-shot since the examples used for further fine-tuning are non-private from the test domain. The scale of noise $C\sigma$ determines the privacy budget in DP-SGD, where higher privacy noise leads to lower privacy leakage $\epsilon$. Training with the same scale of privacy noise may result in different $\epsilon$ reported in Table 9: Different batch sizes $b$ (sampling ratio $p = b/|\mathcal{E}|$) and numbers of iterations $T$ used in training also impact the privacy accounting in DP-SGD (Balle & Wang, 2018).

# F    ADDITIONAL RESULTS FOR STUDIES IN SECTION 4.2

Fig. 4 (Left) shows the joint effect of learning rate $\eta$ and batch size $b$ for BERT.base over zero-shot relation prediction on AMAZ-cloth under the same privacy parameters: Larger batches and learning rates together lead to good performance (diagonal area) under fixed training steps. This observation aligns with the findings in privately fine-tuning LLMs on standard text data (Li et al., 2021). Fig. 4 (Right) shows the impact of norm clipping threshold $C$ for BERT.base on zero-shot relation prediction over MAG-USA/CHN datasets, while other hyperparameters remain the same. The threshold $C$ does not affect the privacy budget $\epsilon$ but is crucial to the utility of DP models that

Figure 4: Effects of learning rate $\eta$ and batch size $b$ (Left), and clipping norm threshold $C$ (Right) for zero-shot relation prediction with privacy preservation.

requires tuning in practice (Bu et al., 2024). Picking a threshold $C$ larger than the actual gradient norm means that most clipping in Eq. (3) is ineffective, and the scale of noise $\sigma C$ is added more than necessary. For instance, $C = 100$ always performs the worst in Fig. 4 (Right). In general, small values of $C$ work better for relational learning as suggested in the general practice and observation of DP learning over non-relational data (Tramer & Boneh, 2021; Li et al., 2021).

