# OpenReview forum: "Privately Learning from Graphs with Applications in Fine-tuning Large Pretrained Models"
_ICLR.cc/2025/Workshop/BuildingTrust — BuildingTrust_

### Official Review · Reviewer_MUU5 · 2025-03-02
**The paper identifies the problem of applying differential privacy in relation learning especially DP-SGD. Provide a novel privacy-preserving pipeline for fine-tuning LLM as proof-of-concept.**

**Rating:** 8
**Confidence:** 3

**Review:**

Strengths:
1. The paper is well-written and easy to follow
2. The problem statement of applying differential privacy in graphs has broad impact one of them being fine-tuning LLM in relational learning

Weakness and errors:
1. Typo error on line 255 d,p are the input and output dimensions not d,q
2. The novelty of the paper can be improved upon negative sampling and efficient per-tuple gradient estimation

Questions:
1. Experiments using GNN based methods are missing
2. Experiments on large scale homophilic and heterophilic graph datasets

---

### Official Review · Reviewer_XiyP · 2025-03-02
**Decent work on differentially private relational learning**

**Rating:** 6
**Confidence:** 3

**Review:**

### Summary

This paper addresses the problem of differentially private learning for relational data. Since DP-SGD relies on gradient decoupling, it is not directly applicable, and the problem addressed here is relevant. The authors approach this problem by introducing a method to decouple gradients by sampling relations.


### Strengths

1. This paper addresses a relevant problem, since differentially private learning poses several challenges in relational learning.
2. Nicely written. The organization of the paper is generally good and the flow is easy to follow.
3. Technical quality of this work is decent.
4. The figure 1 is quite instructive and helps understand the approach better.

### Weaknesses

1. The formulation of this work is not very novel. Negative sampling methods have already been explored and this paper combines it with DPSGD to show its use in privacy preservation.
2. A rather interesting point from Table 8 is that the performance of the model with $\varepsilon=4$ outperforms the non-private model. This result merits discussion but is not discussed in the paper.

---

### Decision · Program_Chairs · 2025-03-04

Accept